# Rotation-Invariant Local-to-Global Representation Learning for 3D Point Cloud

**Seohyun Kim**      **Jaeyoo Park**      **Bohyung Han**
Computer Vision Laboratory & ASRI, Seoul National University
{goodbye61, bellos1203, bhhan}@snu.ac.kr

## Abstract

We propose a local-to-global representation learning algorithm for 3D point cloud data, which is appropriate for handling various geometric transformations, especially rotation, without explicit data augmentation with respect to the transformations. Our model takes advantage of multi-level abstraction based on graph convolutional neural networks, which constructs a descriptor hierarchy to encode rotation-invariant shape information of an input object in a bottom-up manner. The descriptors in each level are obtained from a neural network based on a graph via stochastic sampling of 3D points, which is effective in making the learned representations robust to the variations of input data. The proposed algorithm presents the state-of-the-art performance on the rotation-augmented 3D object recognition and segmentation benchmarks. We further analyze its characteristics through comprehensive ablative experiments.

## 1   Introduction

3D object recognition based on point cloud data has witnessed remarkable achievement in recent years thanks to deep learning technologies [1–10]. While sparse and irregular structures and missing and noisy information of 3D point cloud data have been addressed actively using deep neural networks, their geometric transformations remain challenging problems; transformation-invariant representation learning requires a more principled formulation. Among many geometric transformations, rotation is particularly challenging to handle in practice, and existing algorithms often rely on the assumption of upright object poses. One of the straightforward solutions to address this issue without the prior is data augmentation, but it is not trivial to cover all possible rotations and generalize on realistic examples due to high computational cost and unexpected corner cases.

There exist a handful of techniques to tackle geometric transformations of 3D point cloud data in the context of 3D object recognition. For example, [1, 2] canonicalize input point coordinates using a spatial transform network, but they require data augmentation to work consistently on the variable transformation of input examples. More recent works [11–14] attempt to employ handcrafted transformation-invariant features such as distances and angles for robust recognition. Although these approaches present practical performance gains, such low-level geometric features have limited capability to express detailed shape information and their computation suffers from an asymptotic increase of computational complexity due to the joint consideration of multiple points.

To address such critical challenges, we introduce a novel rotation-invariant 3D object recognition framework based on graph convolutional neural networks (GCNs). The proposed approach designs the descriptors based on the local reference frame (LRF), *i.e.*, a local coordinate system. The receptive field of our descriptor is enlarged hierarchically and stochastically, which leads to the construction of better regularized and representative features even with substantial variations of the object shape. We utilize GCNs upon stochastically generated graphs and encode local-to-global shape information effectively, which provides the capability to represent global rotation-invariant information of a target

object and facilitates the robustness to perturbations and outliers in observations. The source codes are available on our project page[1].

The contributions of this paper are summarized below:

- We introduce a local-to-global representation learning method for 3D point cloud data based on GCNs, referred to as RI-GCN, to model rotation-invariant features in a progressive manner without computing any geometric features, such as angle or distance.

- RI-GCN enlarges receptive field sizes and regularizes learned features by computing the descriptors stochastically. This strategy is effective in enhancing robustness to perturbations and outliers.

- RI-GCN presents great performance gain on 3D object classification and segmentation benchmarks based on point cloud data, even without data augmentation with respect to rotation.

The rest of the paper is organized as follows. We first discuss existing works about object recognition based on 3D point cloud data in Section 2. Section 3 describes the proposed approach to generate rotation-invariant local descriptors and construct graph convolutional neural networks using the features hierarchically. We present the experimental results on the standard benchmarks in Section 4, and make the conclusion in Section 5.

## 2  Related Works

**Deep learning for 3D point cloud recognition**   With the remarkable advancement of deep neural networks, the task of recognizing 3D objects based on the raw coordinates of inputs has also received a lot of attention. As a pioneering work, PointNet [1] has been proposed to extract global features from the unordered set of original points. However, PointNet lacks the ability to understand the local features within an object, which play a crucial role in various tasks related to 3D object recognition. Since then, strategies for learning local features, such as incorporating edge information [2] and building hierarchical models [3, 4], have provided impressive performance gains in the classification task. However, they still suffer from performance degradation induced by rotated inputs. In our work, we aim to construct a novel 3D object recognition framework, which is robust to rotation and achieves outstanding performance.

**Rotation-robust learning for 3D point cloud recognition**   Designing rotation-robust feature representations is critical to improve the overall accuracy in 3D object recognition. Initially, spatial transform networks (STNs) [1, 2] have been employed to provide robustness against rigid geometric transformations. Then, rotation-equivariant networks [15, 16] based on spherical functions have been proposed to improve robustness to the rotation. SFCNN [17] presents an approach to transform the 3D point cloud into an icosahedral lattice. However, all of the aforementioned methods are still vulnerable to unseen orientations and rely heavily on rotation-augmented data to recognize objects successfully. More recent methods often extract geometric features such as distances and angles by considering multiple points jointly; [13] employs the relative angle of normal vectors and the difference vector of point pairs while [11] aggregates the information of the azimuthal/polar angle and radial distance between two points. Similarly, [12] utilizes the distances and angles within local triangular structures, which is built upon a reference point, a local neighborhood centroid, and local neighborhood points. However, the manual feature extraction steps in [11–13] may lead to critical information loss, which induces ambiguities in recognizing objects. On the contrary, we learn rotation-invariant local descriptors based on LRFs to encode local shape information, and aggregate those local descriptors hierarchically to obtain global features. Consequently, our model learns local features directly from the original 3D points, which is more effective in maintaining the information in the original data, using both shallow and deep learning algorithms, *e.g.,* PCA and MLP.

**Graph-based networks for 3D point cloud data**   Graph convolutional neural networks (GCNs) have made a lot of progress in the past decade [18–21] and achieved great success on many machine learning tasks. The graph-based approaches are often categorized into two groups: 1) spatial structure analysis techniques and 2) methods based on spectral graph theory. In general, the approaches based

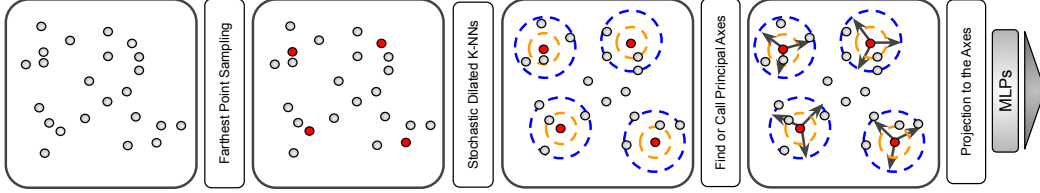

Figure 1: Illustration of descriptor extraction module. Given a set of points, we select a set of representative points (in red) using the FPS algorithm. Then, we use the stochastic dilated $k$-NNs (orange & blue circles) for searching neighbors. Each local region finds the principal axes for rotations of local points. Then, the transformed regions are mapped to high dimensional features using MLPs.

on the spatial graphs set their focus on edge information [2, 22, 23]. DGCNN [2] has proposed an edge convolution operation; it constructs a graph in every layer dynamically to extract local geometric features. ECC [22] presents an edge-conditioned convolutional network, which generates an edge-specific weight matrix and aggregates neighborhood features spatially. DeepGCNs [23] claims that the spatial graph convolutional layers can be stacked deeply using the residual connections while mitigating vanishing gradient and over-fitting issues. On the other hand, the approaches based on spectral graph theory [18–20] have been studied concurrently. [7] constructs a graph structure on the whole 3D point cloud inputs and performs spectral graph filtering using the filters approximated by Chebyshev polynomials [19]. Instead of applying graph convolution to the entire point cloud, [4] leverages spectral graph filtering [20] to compute local descriptors with a hierarchical structure, which leads to better performance than the techniques relying on pure multi-layer perceptrons. However, [4, 7] are still vulnerable to geometric transformations since the graph signal is represented by the raw 3D coordinates. In contrast, we apply an approximate spectral graph filtering technique [20] to the local point features that are generated in a rotation-invariant manner.

## 3 Learning Stochastic Rotation-Invariant Representations

### 3.1 Overview

Our goal is to build a rotation-invariant 3D object recognition framework, which takes an unordered set of point clouds $\mathcal{P} = \{p_1, \ldots, p_N\}$ as input, where $p_i \in \mathbb{R}^3$ $(i = 1 \ldots N)$. In other words, we aim to learn a differentiable function $f : \mathbb{R}^{N \times 3} \to \mathbb{R}^F$, which satisfies the following property:

$$f(\mathcal{P}) = f(r(\mathcal{P})), \tag{1}$$

where $r : \mathbb{R}^{N \times 3} \to \mathbb{R}^{N \times 3}$ is an arbitrary SO(3) rotation mapping function.

Our framework, RI-GCN, consists of three modules, *descriptor extraction, descriptor extension* and *graph-based abstraction*. The first one, descriptor extraction, constructs local shape descriptors by stochastically sampling representative points and encoding the points upon a corresponding local reference frame using a neural network model. Next, the descriptor extension module expands the scope of each descriptor in a progressive manner and organizes local-to-global representation hierarchies. Finally, in the graph-based abstraction step, GCNs are employed to acquire context-aware feature representations by incorporating neighboring descriptors.

### 3.2 Descriptor Extraction

This module learns a set of local descriptors based on representative points and their neighboring points. Each representative point and its neighbors are projected onto a local reference frame (LRF) given by the principal component analysis estimated in the local region. This idea is motivated by the observation that the descriptors in a local region are not affected by the rotations with respect to the global coordinate system. Figure 1 illustrates an overview of this module.

The first step of the module is to identify representative points and locate their neighbors. We select a set of representative points $\mathcal{Q} = \{q_1, \ldots, q_M\} \in \mathbb{R}^{M \times 3} \subset \mathcal{P}$ $(M \ll N)$ using the farthest points

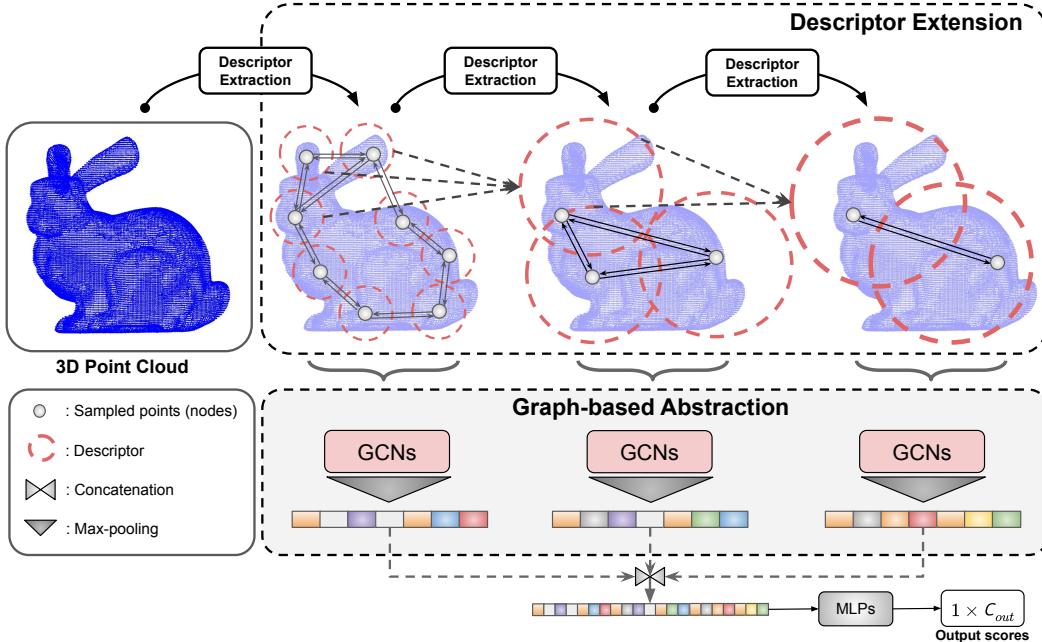

Figure 2: The procedure of RI-GCN for rotation-invariant 3D object classification. The descriptor extension step comprises multiple stacks of the descriptor extraction module, which expands the scope of descriptors by grouping the local features while maintaining rotation invariance. The graph-based abstraction module aggregates local descriptors using graph convolutional neural networks that model the topological structure of the descriptors at each hierarchy.

sampling (FPS) algorithm [24]. Formally, the objective function of the FPS algorithm is

$$\max_{p \in \mathcal{P}} \min_{q \in \mathcal{Q}} \text{dist}(p, q), \tag{2}$$

where $\text{dist}(\cdot, \cdot)$ is a metric distance function. Although the problem of identifying the optimal $\mathcal{Q}$ is NP-hard, we can approximate the solution by iteratively finding the farthest point at every stage in a greedy manner until the number of the selected points reaches $M$. This solution is known as a 2-approximation algorithm.

Each representative point $q_i$ $(i = 1 \ldots M)$ defines a local region by searching for its neighbors from the original point set $\mathcal{P}$. To handle large variations in shape, we employ a stochastic version of the dilated $k$-NN search [23]. Given an anchor point $q_i$ and a dilation rate $d$, the output of the dilated $k$-NN search is an index set of $k$ points in $\mathcal{P}$, which are located at every $d$-th position in the sorted list of the distances from $q_i$. We add stochasticity to the dilated $k$-NN search by setting both $k$ and $d$ as random variables during training, which has strong regularization effects and improves the performance of our approach.

Next, we construct a local reference frame based on each $q_i$ and its neighbors. As shown in Figure 1, we adopt two local point sets, $\mathcal{N}_{k,1}(q_i)$ (orange circle) and $\mathcal{N}_{k,d}(q_i)$ (blue circle), with respect to each anchor point $q_i$ using two different search scopes. Note that the dilation rate of $\mathcal{N}_{k,1}(q_i)$ is set to 1 while that of $\mathcal{N}_{k,d}(q_i)$ is a random variable $d$. The local reference frame is obtained by principal component analysis using the points in $\mathcal{N}_{k,d}(q_i)$, which are then projected onto the estimated reference frame. We denote the projected set of points as $\mathcal{N}'_{k,d}(q_i)$.

The last step of this module generates the local descriptors based on the rotation-normalized information in the local reference frame. The descriptor corresponding to $q_i$, denoted by $\phi_i \in \mathbb{R}^C$, is obtained by

$$\phi_i = f_\Theta([g_1(\mathcal{N}'_{k,1}(q_i)), g_2(\mathcal{N}'_{k,d}(q_i))]), \tag{3}$$

where $f_\Theta(\cdot)$, $g_1(\cdot)$ and $g_2(\cdot)$ are multi-layer perceptrons followed by max pooling layers, and $[\cdot, \cdot]$ denotes tensor concatenation in the channel direction.

### 3.3 Descriptor Extension

Suppose that we are given a set of points and their descriptors, $\mathcal{Q} = \{q_1, \ldots, q_M\}$ and $\mathcal{H} = \{\phi_1, \ldots, \phi_M\}$, by the descriptor extraction module. Then, the first level of the descriptor extension module takes those two sets, $\mathcal{Q}_0 \equiv \mathcal{Q}$ and $\mathcal{H}_0 \equiv \mathcal{H}$, as its inputs and produces the same type of sets for the input of the subsequent level, $\mathcal{Q}_1$ and $\mathcal{H}_1$, where $|\mathcal{Q}_1| \ll |\mathcal{Q}_0|$. The descriptor extension module repeats this procedure for a predefined number of iterations.

Basically, the procedure in the descriptor extension module is almost identical to the descriptor extraction module, but they have the following differences. First, each level of this module deals with a reduced number of points by applying an additional farthest point sampling procedure, which leads to updating local point sets. Second, in each level of this module, $g_2(\mathcal{N}'_{k,d}(q_i))$ in Eq. (3) is replaced by the descriptor representation obtained from the previous level, e.g., $\phi_i$. Third, we reuse the principal axes obtained from the descriptor extraction module, rather than recompute them in each level. That is because, as the hierarchy goes up, points become sparser and the newly obtained axes would not be stable.

In a nutshell, this module constructs a hierarchical structure to extend the coverage of the local descriptors obtained from the descriptor extraction module and elevates the semantic levels of the descriptors with the extended scopes. This is achieved by reducing the number of points and aggregating the information in nearby features, which facilitates local geometry understanding over a wider area and saves computational cost in a progressive fashion. Note that it is highly desirable to enlarge the scope of the descriptors since local shapes can be identified and characterized effectively with large receptive fields. Figure 2 illustrates the pipeline of our description extension module.

### 3.4 Graph-based Abstraction

The objective of the graph-based abstraction module is to obtain context-aware descriptors by referring to the neighborhood information and achieve the representations robust to noise and outliers. Since the information learned at each hierarchy so far is limited to modeling the partial shape of an object, it is desirable for each descriptor to be aware of its geometric context to understand the overall shape. To aggregate the information of the descriptors, we apply a GCN at the end of each level, where the graphs are constructed stochastically to cover the various scopes and learn diverse contexts.

Specifically, the descriptors at each level are first converted to the graph signal $X_l \in \mathbb{R}^{n_l \times c_l}$, where $n_l$ is the number of nodes and $c_l$ is the feature dimension at the $l$-th hierarchy. For simplicity, we omit the index $l$ for the rest of this section. For the adjacency matrix $A \in \mathbb{R}^{n \times n}$, we construct a $k$-NN graph while setting the number of edges to the neighbors as a random variable $\hat{k}$. We set the weights of the edges between nodes using the Euclidean distances smoothed by a Gaussian kernel.

Based on the constructed graph, we adopt a GCN following [20], which introduces a renormalization trick to approximate the graph as $\hat{A} = \tilde{D}^{-\frac{1}{2}} \tilde{A} \tilde{D}^{-\frac{1}{2}}$, where $\tilde{A} = A + I_n$ is the adjacency matrix with self-connections, and $\tilde{D}_{ii} = \sum_j \tilde{A}_{ij}$ is a degree matrix. Then, a GCN layer is applied to enhance the representational power of $X$ as

$$Y = \text{ReLU}\left(\hat{A}XW\right), \tag{4}$$

where $W$ is a learnable parameter matrix. As illustrated in Figure 2, the outputs from all of the hierarchies are max-pooled and are concatenated to obtain the final representation for classification.

## 4  Experiments

This section presents the experimental results of our algorithm compared to existing approaches. We demonstrate the effectiveness of our framework via several ablation studies. The evaluation protocol in this paper is identical to those conducted in previous works [11, 12, 16, 17]. Specifically, we perform experiments in three modes: training and testing with azimuthal rotations (z/z), training with azimuthal rotations and testing with arbitrary rotations (z/SO(3)), and training and testing with arbitrary rotations (SO(3)/SO(3)).

Table 1: 3D object classification results on ModelNet40. The last column, Drop, shows the accuracy difference between SO(3)/SO(3) and $z$/SO(3).

| Method | Input (dimensionality) | $z$/$z$ (%) | $z$/SO(3) (%) | SO(3)/SO(3) (%) | Drop |
|---|---|---|---|---|---|
| PointNet (w/o STN) [1] | pc (1024 x 3) | 88.5 | 16.4 | 70.5 | 54.1 |
| PointNet++ (MSG w/o STN) [3] | pc+normal (5000 x 6) | 91.9 | 18.4 | 74.7 | 56.3 |
| SO-Net (w/o STN) [25] | pc+normal (5000 x 6) | **93.4** | 19.6 | 78.1 | 58.5 |
| DGCNN (w/o STN) [2] | pc (1024 x 3) | 91.2 | 16.2 | 75.3 | 59.1 |
| PointNet [1] | pc (1024 x 3) | 89.2 | 16.2 | 75.5 | 59.3 |
| PointNet++ [3] | pc+normal (5000 x 6) | 91.8 | 18.4 | 77.4 | 59.0 |
| SO-Net [25] | pc+normal (5000 x 6) | 91.2 | 21.1 | 80.2 | 59.1 |
| DGCNN [2] | pc (1024 x 3) | 92.2 | 20.6 | 81.1 | 60.5 |
| SpecGCN [4] | pc (1024 x 3) | 91.5 | 28.8 | 75.3 | 46.5 |
| Spherical CNN [16] | Voxel (2 x 64 x 64) | 88.9 | 76.9 | 86.9 | 10.0 |
| SFCNN [17] | pc (1024 x 3) | 91.4 | 84.8 | 90.1 | 5.3 |
| SFCNN [17] | pc+normal (1024 x 6) | 92.3 | 85.3 | **91.0** | 5.7 |
| RIConv [12] | pc (1024 x 3) | 86.5 | 86.4 | 86.4 | **0.0** |
| ClusterNet [11] | pc(1024 x 3) | 87.1 | 87.1 | 87.1 | **0.0** |
| RI-GCN (ours) | pc (1024 x 3) | 89.5 | 89.5 | 89.5 | **0.0** |
| RI-GCN with normals (ours) | pc+normal (1024 x 6) | 91.0 | **91.0** | **91.0** | **0.0** |

## 4.1 3D Object Classification

To evaluate the robustness to rotation, we compare the proposed algorithm, RI-GCN, with recent 3D object classification approaches on ModelNet40 [26], a widely used benchmark. It consists of CAD models in 40 categories and contains 9,843 and 2,468 shapes for training and testing, respectively. Point clouds are sampled uniformly from vertices and faces of the CAD models. All points are shifted and normalized to fit in a unit sphere.

Table 1 presents the overall performance of 3D object classification techniques in the three modes, including two recent rotation-invariant algorithms [11, 12]. Most approaches [1–3, 16, 25] show great performance in $z$/$z$. However, it is noticeable that their performance is significantly degraded in the presence of unseen rotations for inference. Moreover, although arbitrary rotations are employed for data augmentation during the training phase in SO(3)/SO(3), the generalization performance is not fully recovered in most algorithms. Although PointNet [1], PointNet++ [3], SO-Net [25], and DGCNN [2] incorporate a spatial transformer network (STN) to improve accuracy, it turns out that STN is not effective in handling the arbitrary rotations, SO(3), regardless of data augmentation. SpecGCN [4], which utilizes spectral graph filtering to construct local descriptor, is not robust to rotation since the graph signal is based on absolute 3D coordinates. Meanwhile, although SFCNN [17] achieves competitive accuracy for the rotations exposed during training via data augmentation, it suffers from handling unseen types of rotations, $z$/SO(3), and presents large gaps compared to the accuracies in SO(3)/SO(3). The existing techniques that address rotation invariance explicitly [11, 12] maintain their recognition accuracies for unseen types of rotations, but their baseline performance is not satisfactory. Our algorithm demonstrates the state-of-the-art accuracy consistently for all the cases and the robustness to unseen types of rotations.

## 4.2 Ablation Study

We perform several ablation studies on ModelNet40 to analyze the effectiveness of our approach.

**Benefit of stochastic learning** To show the effectiveness of stochastic learning, we analyze the impact of three stochastic factors to learn the descriptors—dilation rate $d$, number of nearest neighbors $k$, and number of edges for the construction of a graph $\hat{k}$—and compare the results with the deterministic approach. Table 2 presents that the model from the pure deterministic learning has the lowest accuracy and adding stochastic learning components improves results consistently.

**Graph convolutional networks (GCNs) vs. multi-layer perceptrons (MLPs)** Table 3(a) presents the comparison between GCNs and MLPs as recognition models. For comparison, we

Table 2: Contribution of stochastic learning. The checkmark indicates that the parameter is given stochasticity. The parameters, $d$, $k$, and $\hat{k}$, denote dilation rate, number of neighbors in the stochastic dilated $k$-NNs, and number of edges for each node in the graphs, respectively.

| | $d$ | $k$ | $\hat{k}$ | $z$/SO(3) (%) |
|---|---|---|---|---|
| Deterministic | - | - | - | 87.7 |
| | ✓ | - | - | 88.6 |
| | - | ✓ | - | 88.9 |
| | - | - | ✓ | 88.0 |
| Partially stochastic | ✓ | ✓ | - | 89.2 |
| | ✓ | - | ✓ | 89.0 |
| | - | ✓ | ✓ | 88.9 |
| Fully stochastic | ✓ | ✓ | ✓ | **89.5** |

Table 3: 3D object classification accuracies on ModelNet40 with the variations of our model.

| Ablation types | Variations | $z$/SO(3) (%) | SO(3)/SO(3) (%) |
|---|---|---|---|
| (a) Architectural variations | MLP | 89.1 | 89.2 |
| | GCN (ours) | **89.5** | **89.5** |
| (b) Transformation scopes | Global transformation | 87.2 | 87.2 |
| | Local transformations (ours) | **89.5** | **89.5** |
| (c) Levels of models | Single level | 88.7 | 88.9 |
| | Two levels | 89.1 | 89.3 |
| | Three levels (ours) | **89.5** | **89.5** |
| | Four levels | 89.3 | 89.4 |
| (d) Stochastic dilation | Deterministic dilated $k$-NN [23] | 88.3 | 88.3 |
| | Stochastic dilated $k$-NN (ours) | **89.5** | **89.5** |

replace a single GCN layer with a single fully-connected layer in all hierarchies, where the dimension of the learnable parameter is consistent with GCNs. GCNs presents better performance than MLPs, which is mainly because GCNs consider the neighborhood of individual nodes and capture their geometric context more effectively.

**Comparison between local and global transformations**   We compare the feature learning based on local transformations with the universal global rotation. To this end, we train a new model identical to ours, except that all 3D points in $\mathcal{P}$ are transformed by a global rotation matrix estimated by the principal component analysis. All the hyperparameters for network configuration and model training are identical to our algorithm based on local transformations. Table 3(b) presents that the proposed method based on local coordinate systems outperforms the model based on the global transformation.

**Effectiveness of hierarchical modeling**   To demonstrate the effectiveness of the proposed hierarchical representations, we evaluate classification accuracy with respect to the number of levels in our neural network architecture. Table 3(c) presents that enlarging the receptive field of a descriptor using multiple levels improves accuracy gradually while adding more than three levels is not particularly helpful.

**Impact of stochastic dilation in $k$-NN search**   Contrary to [23], the proposed algorithm employs a stochastic dilated $k$-NN search by setting $k$ and $d$ as random variables. In this way, the scope of the nearest neighbor search, $k \times d$, is determined stochastically, which regularizes the local descriptors and make the learned model robust to noise and shape variations. Table 3(d) clearly illustrates the effectiveness of stochastic dilation in our nearest neighbor search strategy.

### 4.3   Robustness to Noise and Outliers

We also conduct experiments for evaluating robustness to noise in input data and outliers. To this end, each input point is perturbed by adding noise from zero-mean Gaussian distributions, $N(0, \sigma^2)$ while

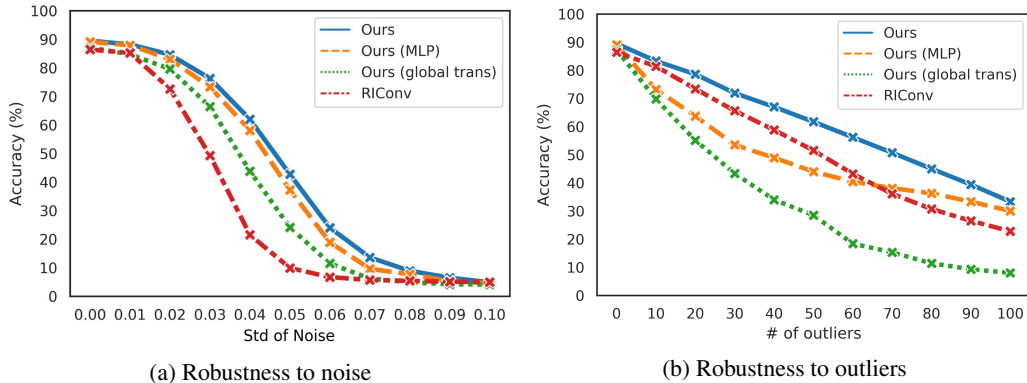

| | | | | | |
|---|---|---|---|---|---|
| (a) Robustness to noise | | | (b) Robustness to outliers | | |

Figure 3: Robustness of our algorithm to noise and outliers on ModelNet40. The results in (a) are obtained by adding Gaussian noise to each point with a different standard deviation while the outliers sampled from the unit sphere are injected for the experiment (b). Note that the model with global transformations is trained with the same protocol as the proposed algorithm.

Table 4: Qualitative results on the rotated inputs in ModelNet40. All the frameworks are trained with data augmentation only for the azimuthal rotation ($z$/SO(3)).

| Method | | | | | |
|---|---|---|---|---|---|
| GT | **airplane** | **table** | **bowl** | **car** | **xbox** |
| PointNet [1] | plant | stairs | plant | stairs | tv-stand |
| PointNet++ [3] | plant | door | vase | bottle | tent |
| DGCNN [2] | plant | stool | vase | plant | table |
| SpecGCN [4] | plant | chair | plant | bottle | tent |
| RIConv [12] | **airplane** | **table** | tent | stairs | range-hood |
| RI-GCN (ours) | **airplane** | **table** | **bowl** | **car** | radio |

outliers sampled from a unit sphere are injected to existing points for each object instance. Note that each object is composed of 1,024 points in all compared algorithms.

Figure 3(a) presents the curves with respect to the level of perturbation, which shows the robustness to perturbation of our algorithm based on GCNs. The use of MLPs instead of GCNs degrades performance marginally in the whole perturbation levels while the global transformation is not effective compared to the local counterpart. One of the state-of-the-art rotation-invariant approaches, RIConv [12], suffers from a substantial accuracy drop for this kind of noise type.

Figure 3(b) illustrates the accuracy of all the compared methods in the presence of outliers, where our full algorithm shows outstanding performance. In particular, GCNs achieves significant accuracy gains compared to MLPs. Note that the local transformations are significantly better than the global one, which implies that outlier injection distorts global shapes and orientations critically. The results from RIConv [12] indicate that the representation learning with the lower-level feature is not robust enough to perturbation noise as well as outliers.

## 4.4 Qualititative Results

Table 4 demonstrates qualitative results for the classification task based on 3D point cloud data. We present the predictions of other models [1–4, 12] for the rotated inputs sampled from 5 object classes—airplane, table, bowl, car, xbox—in ModelNet40. All algorithms are trained with data augmentation for the azimuthal rotation ($z$/SO(3)). The methods equipped with STN including PointNet [1], PointNet++ [3], and DGCNN [2] are not good at handling rotated inputs in all the presented cases, and SpecGCN [4] is not particularly better. Although RIConv [12] successfully recognizes rotated objects with salient features such as wings of an airplane and legs of a table, it fails

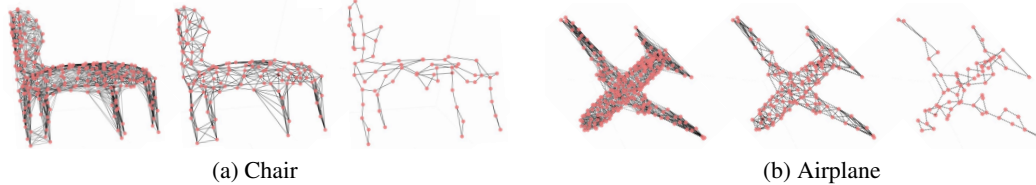

|      (a) Chair      |      (b) Airplane      |

Figure 4: Qualitative examples of generated graphs in all the three levels. Circles in red indicate nodes and solid lines denote edges between nodes.

Table 5: 3D part segmentation results on ShapeNet. The last column, Drop, denotes the performance degradation from SO(3)/SO(3) to $z$/SO(3). The results of other methods are from [12].

| Method | Input (dimensionality) | $z$/SO(3) (%) | SO(3)/SO(3) (%) | Drop |
|---|---|---|---|---|
| PointNet [1] | pc (1024 x 3) | 37.8 | 74.4 | 36.6 |
| PointNet++ [3] | pc+normal (5000 x 6) | 48.2 | 76.7 | 28.5 |
| DGCNN [2] | pc (1024 x 3) | 37.4 | 73.3 | 35.9 |
| RIConv [12] | pc (1024 x 3) | 75.3 | 75.5 | 0.2 |
| RI-GCN (ours) | pc (1024 x 3) | **77.2** | **77.3** | **0.1** |

to identify objects with simpler structures, including bowl and car. Note that the proposed algorithm denoted by RI-GCN identifies bowl and car successfully, which is partly because it captures global shapes more effectively by exploiting multi-level GCNs. Our approach fails in predicting Xbox correctly, but its shape is too indistinguishable to be recognized even by a human.

Figure 4 presents the examples of graphs that are generated along the hierarchies. The nodes in the graph at the first level are densely connected and the graph becomes sparser as the level goes up. The overall skeleton of the object is captured more effectively at the top of the hierarchy, which helps extract global representation. Actually, the learned representations from the GCNs at multiple levels are complementary to one another.

## 4.5  3D Part Segmentation

We conduct an additional experiment of 3D part segmentation following [12] to verify the rotation invariance of the proposed approach. The objective of this task is to predict a part label for each point of an object. We employ the ShapeNet dataset [27], which consists of 16,881 shapes from 16 object categories with 50 part labels in total. We adopt the hierarchical point feature propagation strategy proposed by PointNet++ [3] to obtain upsampled feature maps, which provides all the original points with their own feature descriptors for part predictions.

Table 5 illustrates the overall performance of 3D part segmentation methods in the $z$/SO(3) and SO(3)/SO(3) modes. The results imply that RI-GCN is also robust to arbitrary rotations of points for the 3D part segmentation task and outperforms the state-of-the-art approaches, especially in the $z$/SO(3) scenario.

## 5  Conclusion

We proposed a novel framework for rotation-invariant recognition with 3D point cloud data, referred to as RI-GCN, which extracts rotation-invariant local descriptors based on local reference frames and employs graph convolutional neural networks to aggregate the local features and learn their topological structures. The proposed stochastic learning strategy regularizes the geometric transformations, especially, rotations, of 3D objects and improves recognition accuracy substantially even in the presence of perturbations and noise. Our approach is directly applied to raw point cloud data without any conversion to low-level handcrafted features, which is unique among the rotation-invariant 3D object recognition techniques based on the 3D point cloud. The graph convolutional neural networks successfully learn global descriptors by combining neighboring local features and capturing hierarchical structures, leading to state-of-the-art performance on rotation-augmented 3D object classification and segmentation benchmarks.

## Broader Impact

The major goal of our research is to enhance the 3D object recognition by imposing the capability to handle geometric transformation, especially, rotations, effectively. 3D object recognition is applicable to various computer vision systems including autonomous driving, visual surveillance, augmented reality, and many others. From the societal point of view, the improvement of 3D object recognition algorithms has both bright and dark sides. These systems can benefit those who are visually impaired by providing assistive vision systems including smart glasses. Meanwhile, the technology might infringe on personal privacy if equipped in drones and CCTVs.

## Acknowledgments

This work was partly supported by Samsung Research Funding Center of Samsung Electronics under Project Number SRFC-IT1801-10 and Institute for Information & Communications Technology Promotion (IITP) grant funded by the Korea government (MSIT) [2017-0-01779, 2017-0-01780].

## Footnotes

[1]https://cvlab.snu.ac.kr/research/rotation_invariant_l2g/

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
