[Supplementary Material]

# Rotation-Invariant Local-to-Global Representation Learning for 3D Point Cloud
## *Supplementary Document*

**Seohyun Kim**  **Jaeyoo Park**  **Bohyung Han**
Computer Vision Laboratory & ASRI, Seoul National University, Korea
`{goodbye61, bellos1203, bhhan}@snu.ac.kr`

## 1  Implementation Details

Our model is implemented in Tensorflow. For the 3D object classification experiments, the learning rate is 0.001 and the batch size is 32. We downsample the original point cloud data to 256, 128, and 64 points in each descriptor extraction module in our framework. In each hierarchy, the number of neighbors for the stochastic dilated $k$-NNs, denoted by $k$, is sampled from the following three different uniform distributions: $\mathcal{U}(32, 96)$, $\mathcal{U}(16, 48)$, and $\mathcal{U}(8, 24)$. The dilation rate employed in the first descriptor extraction stage is sampled from $\mathcal{U}(2, 4)$. Meanwhile, the number of edges in the graph, $\hat{k}$, is given by sampling from $\mathcal{U}(8, 24)$, $\mathcal{U}(4, 12)$, and $\mathcal{U}(2, 8)$. All the parameters are fixed throughout the experiment, while $k$ and $\hat{k}$ are set to $(64, 32, 16)$ and $(16, 8, 4)$, respectively, along the hierarchies for inference. The dilation rate, $d$, is 3. The experiment for 3D object classification was conducted on a single NVIDIA Titan XP GPU.

The inference of our algorithm takes 0.56 ms for a single point cloud object. PCAs spend most of the run-time, which is partly because they are performed on CPUs, and it would be possible to improve speed by parallelizing the computations on GPUs.

For the 3D part segmentation experiments, the learning rate is 0.001 and the batch size is 50. We downsample to 512, 128, and 32 points in each descriptor extraction step in our framework. Similar to the classification task, $d$ is set to 3 and $k$ is sampled from $\mathcal{U}(56, 72)$, $\mathcal{U}(56, 72)$, and $\mathcal{U}(8, 24)$ while $\hat{k}$, is determined by sampling from $\mathcal{U}(56, 72)$, $\mathcal{U}(28, 36)$, and $\mathcal{U}(4, 12)$. The dilation rate employed in the first descriptor extraction stage is sampled from $\mathcal{U}(2, 4)$. Note that we search for the same number of neighbors in the first two hierarchies by following the previous works [2, 4]. We fix all the parameters for the experiment, and set $k$ and $\hat{k}$ to $(64, 64, 16)$ and $(64, 32, 8)$, respectively, along the hierarchies for inference. The dilation rate, $d$, is 3. The 3D part segmentation experiments are performed on a single NVIDIA Tesla V100 GPU.

## 2  Visualization of Local Reference Frames

Figure 1 visualizes the local axes constructed in the descriptor extraction step. The figures on the left present the original point clouds and the right ones illustrate the local reference frames for the downsampled points. The figures show that the directions of the major axes in most of the local reference frames align fairly well to the directions with the largest global variances in general. Such observations support our claim that using a local reference frame helps the model to recognize the object better.

(a) Chair

(b) Airplane

Figure 1: Visualization of the estimated principal axes for each subsampled point.