[Reviews · NeurIPS 2020]

Review 1

Summary and Contributions: The paper proposes a new representation algorithm for 3d point cloud data that explicitly tackle the invariance of the representation to rotations. The method is novel and compared against baselines showing a minor but quantifiable improvement. The reason for my moderate score is that the idea resembles similar ideas used for meshes (but the important difference is that it is applied to point clouds), and the final results are more or less comparable to the state of the art.

Strengths: The idea is simple and seems effective. The authors propose to build a local reference frame using PCA, and then define the descriptors in this reference frame.

Weaknesses: The improvement over baselines is quantifiable, but minor. What is the variance on the final score for different training rounds? Such a small difference could be the result of hyperparameter tuning or random initialization.

Correctness: I believe that the method works but I do not fully understand why. Consider the simple case of a point cloud of a 2D square: in this case, no matter what point you pick, the PCA will only give you one reliable vector (the normal) and the other two are going to be parallel to the square but will be picked randomly depending on the PCA algorithm used. How can this be a good discriminator and why does it make it rotation invariant? The method shares similarities to older works that are trying to similarly pick reference frames on triangular meshes https://arxiv.org/pdf/1501.06297.pdf

Clarity: Good

Relation to Prior Work: Missing citation/discussion to a relevant work: https://arxiv.org/abs/1809.05910

Reproducibility: Yes

Additional Feedback: Updated review: Thank you for the clarification on the PCA. The new test on ShapeNet makes a stronger case for the practical value of the technique, but the novelty is still limited. I am keeping my positive score since I think the technique has a high practical value and tackles an important and timely problem.


Review 2

Summary and Contributions: The paper introduces a representation for 3D point cloud classification task that is rotation-invariant. The proposed algorithm first clusters point cloud in a coarse-to-fine manner, aligns the features to the principal axis, then learns a graph convolution network to reason the contextual relationship. The network is shown to be robust for perturbations as well as rotations, and achieves great performance on 3D classification benchmarks.

Strengths: The paper handles a very crutial problem in 3D understanding, and provides a nice simple solution. The experimental result section is very extensive, and contain many ablation studies as well as visualizations to help justify the proposed approach. I especially like the visualizations of the graph structure, and not to mention the improved robustness for rotations.

Weaknesses: - Although there are several great points in this paper, there has already been several recent works that tackles this problem, as noted in the paper. Could authors extend the description of L75: " the methods above [11–13] lose information during the conversion to low-level features, which induces ambiguities in recognizing an object."? For example, in [12] the lower level features are actually concatenated in the subsequent networks, why it induces ambiguities? - Designing rotation-invariant features is very important, but just demonstrating the approach on a less challenging 3D classification task seems less interesting. Most algorithms already achieve great performances, so the improvements of this paper is a bit marginal. Again, as an example, [12] evaluated on more challenging 3D segmentation task. - It would be nice to come up with a way to visualize the learned principal axis of this approach to help understand the behavior of the algorithm more.

Correctness: Yes.

Clarity: Yes.

Relation to Prior Work: Yes.

Reproducibility: Yes

Additional Feedback:


Review 3

Summary and Contributions: The authors propose a new method for 3D point cloud classification, specifically targeting improvements to point cloud recognition. The key idea is to build a hierarchy local descriptors based on stochastically sampled subsets of points. "Locality" is achieved by projecting sampled substets onto their principal axes before further processing. The authors show empirical evidence that the method learns an invariant representation (generalizes without drop in accuracy to arbitrary rotations when trained on azimuthal rotations only). Extensive empirical analysis is performed to study the effect of different model design choices.

Strengths: - extensive empirical evaluation of the design decisions behind the proposed approach. For example, Table 3b presents evidence that performing a local transformation based on PCA results in a 2.3% improvement in accuracy for both z/SO(3) and SO(3)/SO(3). - empirical results directly support claims that the learned representation is rotation invariant - the proposed approach shows good relative performance to prior work in terms of performance with arbitrary rotations, and good robustness to noise and outliers (although the noise and outlier analysis is slightly limited)

Weaknesses: - there are certain specific details of the model which are not in the main text (see clarity section) - outlier analysis is limited in that it's a performance comparison with RIConv but not ClusterNet (the closest baseline method), but this is a minor point. - evaluation is limited to one dataset. It would be good to see comparisons on indoor 3D scan datasets, despite the fact that applying the method to those datasets may not precisely show its strengths - given the performance improvements, it is worthwhile to understand the computational cost of the model in comparison to prior work. This analysis is currently lacking.

Correctness: The motivation behind the model design makes sense, and there is extensive empirical evaluation to show the effect on performance. However since certain improvements in accuracy are very slight (e.g. 89.3 and 89.5 for two vs. three levels it would be good to see the outcome of multiple runs).

Clarity: The precise network architecture is not described well in the paper. It would be good to include a clear description of the distinction between the model with GCN/MLP. What exactly do the MLP operations look like in the top half of Fig. 2? It would help to have consistent notation between Fig1, Fig2 and Section 3.2. Otherwise I find the writing to be in good shape.

Relation to Prior Work: I think the authors do a good job discussing the proposed work relative to prior methods. They have covered the important comparison with methods that aim to be robust to rotation through building local features [11] and [12], which is the most similar set of works.

Reproducibility: No

Additional Feedback: ====== post rebuttal response ====== I appreciate the authors' response to my questions in the rebuttal. The increase in computational cost on the order of milliseconds is acceptable, especially given that certain procedures can be sped up with improved implementation. The rebuttal allows for greater confidence in the improvement over prior work. I stand by my original rating.


Review 4

Summary and Contributions: This paper aims to address the problem of point cloud classification. It proposes a networks combines point abstraction level in PointNet++ and GCNs. Specifically, the input points are sampled by FPS to obtain sparser points in deeper layers. Each sampled point is taken as anchor to find a set of neighboring points by dilated k-NN with random dilated factors and neighbor size. For the first level, the obtained k-NN neighbors are used to estimate a local orientation to construct a local reference frame to achieve locally rotation invariance. The points in each level after extracting features are then input into GCNs followed by max-pooling to obtain a global representation. All the global represenations obtained at all levels are concatenated to go through a MLP for classification. Experiments are conducted to test the rotation invariance of the proposed methods and other existing methods. Ablation study also given to show the effectiveness of each component.

Strengths: The paper is well written and with good experimental results. Although the results can not beat the state of the art on classification performance, it achieves the best robust to rotation invariance, for which, the performance of most existing methods will degrade significantly. The key to rotation-invariant representation learning is construct a local referece frame which is rotation invariant. Therefore, the local feature computed under this reference is also robust to rotation. The paper combines many existing techniques in a reasonable way to achieve a good performance for rotation invariant point cloud classification. As I will describe in the weakness part, the components of this paper are mostly from existing works.

Weaknesses: The idea of estimating and relying on local reference frame to achieve rotation invariance has been explored before in similar context, thus might downgrade the novelty of this paper. For example, "A-CNN: Annularly Convolutional Neural Networks on Point Clouds, CVPR'19" uses the local point set to estimate the normal as this paper does, the difference is that A-CNN uses this normal to project 3d points into 2d plane, however, the basic idea of them is both to achieve locally rotation invariance. "Relation-Shape Convolutional Neural Network for Point Cloud Analysis, CVPR'19" mentioned in their experiments about rotation invariance that they construct a local reference frame to achieve rotation invariant representation of local point set which is the same as this paper. The randomized technique is also a common technique in training deep networks for exploring a larger data space or parameter space. The whole hierarchy is identical to PointNet++. For each level, the feature abstraction procedure is also similar to PointNet++, e.g., eq(3). Since the proposed method uses two k-nn neighbors, then it has to concatenates their MLPs together and follows another MLP, this could be a common practice and can not be regarded as a new point proposed by this paper. Using GCNs to extract a better point cloud representation has also been explored. To sum up, the paper combines several existing techniques, but it does lead to a better performance and these techniques are combined in a reasonable way. However, the reviewer think that it is not significantly enough for NeurIPS.

Correctness: In L150, it is unclear why to use descriptor representation obtained in previous to replace g2(N) in Equ(3). This means that in the 2nd and 3rd levels, no dilated k-nn is used, it only concatenates the central features in the previous layer with the extracted features based on neihgbors. This is ok, but just what PointNet++ and other point-based methods did. In equ(3), does N'_{k,1} use the sample local reference frame estimated by N_{k,d}? In L75, it claims that "the methods above [11–13] lose information during the conversion to low-level features", this might be not true. [11] is a rigorously rotation-invariant representation. What is more, if these methods will loss information, how about the proposed method? What is the reason/property making the proposed method does not have this problem?

Clarity: yes

Relation to Prior Work: Many ideas are similar to existing works (see my comments above), it lacks a discussion about how this work is related to them.

Reproducibility: Yes

Additional Feedback: My concerns about the novelty and relations to previous works are not well addressed in the rebuttal. I maintain my rating that the paper is not novel enough but has some practical usage.

[Author Response · NeurIPS 2020]

We appreciate positive and constructive comments and address the main concerns raised by the reviewers below.

**Novelty (All)**   The proposed algorithm is a unique combination of a GCN and a novel rotation-invariant local descriptor for object recognition on 3D point cloud data. (It is rare to use GCN for rotation-invariant 3D point cloud recognition.) We build a hierarchical graph structure on top of the learned rotation-invariant *local* descriptors to extract *global* representations. Our descriptor based on the stochastic local reference frame is more effective in handling rotation transformations than SpecGCN [4] and PointNet++ [3], which do not consider rotation-invariance although they adopt similar hierarchical network structures. Moreover, our GCN models are robust to noise and outliers as illustrated in Figure 3 of the main paper while MLP, which can be regarded as the PointNet++ implementation with our rotation-invariant descriptors, suffers from substantial accuracy loss with the challenges.

**Discussion about RIConv [12] and ClusterNet [11] (R2, R4)**   Both methods start with local feature extraction steps, which require the manual descriptor design using distances, angles, and others while our approach learns local features directly from 3D points based on both shallow and deep learning algorithms, *i.e.,* PCA + MLP, before they are fed into a GCN. Note that our training procedure takes the original 3D points and, consequently, is free from information loss. The manual feature extraction steps in RIConv and ClusterNet may incur the loss and lead to performance degradation. In particular, a local triangle and a descriptor in [12] have many-to-one correspondences because a point is expressed in terms of distances and angles with respect to two reference points, which means the point defines a circular trace.

**Test on another challenging dataset (R2, R3)**   We conduct an additional experiment of part segmentation on ShapeNet. Table 4 presents that the proposed method is also effective to rotation-invariant part segmentation and outperforms the state-of-the-art, especially in the z/SO(3) scenario. The results from other methods are copied from [12].

Table 4: Results on ShapeNet in terms of mIoU.

| Method | Input | SO(3)/SO(3) | z/SO(3) |
|---|---|---|---|
| PointNet | xyz | 74.4 | 37.8 |
| PointNet++ | xyz+normal | 76.7 | 48.2 |
| RIConv | xyz | 75.5 | 75.3 |
| Ours | xyz | **77.3** | **77.2** |

**Missing reference and discussion (R1, R4)**   The main reason for so-called the local reference frame in A-CNN [A1] is not for rotation-invariant recognition but for the definition of a convolution operation with unordered data such as point cloud. Therefore, the accuracy of A-CNN on z/SO(3) is as low as 35.8% according to our experiment based on the official code. It is difficult to discuss RS-CNN [A2] rigorously because the detailed information about the local reference frame is missing. However, RS-CNN extracts hand-crafted features as in RIConv and ClusterNet, and employs additional information, normal vectors, for the extension for the local coordinate system. Moreover, it is not clear if the extension leads to invariance to rotation; although it is tested only for two rotation angles, its accuracy is not impressive at all, especially, considering the very high performance of PointNet++. Both MeshCNN [A3] and Geodesic CNN (G-CNN) [A4] are designed for meshes and their target tasks are different from ours. MeshCNN takes advantage of local hand-crafted features but does not use local reference frames. G-CNN defines a local system of discretized polar coordinates built upon triangular meshes, which is not suitable for the recognition on 3D point cloud data because the surface information is missing in our task.

**Clarification (R1, R4)**   Since there are not many points in the higher level of the graph, the dilated $k$-NN search on the sparse points is not suitable. We compute PCAs at the lowest level only since it is sufficient to project points onto the (most) local reference frame for rotation-invariant recognition. Meanwhile, GCN is effective to learn global structures in a progressive manner. In practice, computing PCAs at every level does not affect the overall accuracy at all. Although PCA is sometimes unstable, the topology of the constructed graph is identical while the learned descriptors on the unstable node are different. Such a moderate discrepancy can be handled by GCN and may have a regularization effect. This can be a reason for the good performance of our algorithm in the presence of noise and outliers.

**Variance of results and computational cost (R1, R3)**   The variance of our accuracy based on 5 trials with random initializations is 0.036, which implies statistical superiority to other methods. Our algorithm requires 0.56 ms for inference while [12] takes 0.22 ms. The difference is mainly because PCAs are computed on CPU many times. It would be possible to parallelize their computations on GPU and improve speed.

**Others [All]**   We will add missing details (including axis visualizations) and release our source code for reproduction if our paper is accepted.

# References

[A1] Komarichev, A., et al.: A-CNN: Annularly convolutional neural networks on point clouds. In CVPR. (2019)

[A2] Liu, Y., et al.: Relation-shape convolutional neural network for point cloud analysis. In CVPR. (2019)

[A3] Hanocka, R., et al.: MeshCNN: A network with an edge. In SIGGRAPH. (2019)

[A4] Masci, J., et al.: Geodesic convolutional neural networks on riemannian manifolds. In ICCV workshops. (2015)


[Meta-Review · NeurIPS 2020]

Post-rebuttal, the reviews were borderline, but leaning positive. The reviewers had concerns about novelty, but thought the paper was a good practical contribution to an important and timely problem. The AC is inclined to agree with the positively-inclined reviewers. The AC urges authors to include experiments from the rebuttal as well as timing information in the camera-ready version of the paper, as this was important for the reviewers' discussions.